# Optimized AAV Vectors for TMC1 Gene Therapy in a Humanized Mouse Model of DFNB7/11

**DOI:** 10.3390/biom12070914

**Published:** 2022-06-29

**Authors:** Irina Marcovich, Nicholas K. Baer, Olga Shubina-Oleinik, Rachel Eclov, Clayton W. Beard, Jeffrey R. Holt

**Affiliations:** 1Department of Otolaryngology & Neurology, Boston Children’s Hospital, Harvard Medical School, Boston, MA 02115, USA; irina.marcovich@childrens.harvard.edu (I.M.); nicholas.baer@childrens.harvard.edu (N.K.B.); volha.shubina-aleinik@childrens.harvard.edu (O.S.-O.); 2Audition Therapeutics (BridgeBio Pharma), Raleigh, NC 27607, USA; racheleclov@gmail.com (R.E.); cb@bridgebio.com (C.W.B.)

**Keywords:** inner ear, hair cell, cochlea, vestibular, utricle, saccule, ampulla, hearing loss, gene therapy, TMC1, TMC2, AAV, AAV9-PHP.B

## Abstract

Gene therapy for genetic hearing loss is an emerging therapeutic modality for hearing restoration. However, the approach has not yet been translated into clinical application. To further develop inner-ear gene therapy, we engineered a novel mouse model bearing a human mutation in the transmembrane channel-1 gene (*Tmc1*) and characterized the auditory phenotype of the mice. TMC1 forms the mechanosensory transduction channel in mice and humans and is necessary for auditory function. We found that mice harboring the equivalent of the human p.N199I mutation (p.N193I) had profound congenital hearing loss due to loss of hair cell sensory transduction. Next, we optimized and screened viral payloads packaged into AAV9-PHP.B capsids. The vectors were injected into the inner ears of *Tmc1^Δ/Δ^* mice and the new humanized *Tmc1*-p.N193I mouse model. Auditory brainstem responses (ABRs), distortion product otoacoustic emissions (DPOAEs), cell survival, and biodistribution were evaluated in the injected mice. We found broad-spectrum, durable recovery of auditory function in *Tmc1*-p.N193I mice injected with AAV9-PHP.B-*CB6-hTMC1-WPRE*. ABR and DPOAE thresholds were equivalent to those of wild-type mice across the entire frequency range. Biodistribution analysis revealed viral DNA/RNA in the contralateral ear, brain, and liver but no overt toxicity. We conclude that the AAV9-PHP.B-*CB6-hTMC1-WPRE* construct may be suitable for further development as a gene therapy reagent for treatment of humans with genetic hearing loss due to recessive *TMC1* mutations.

## 1. Introduction

Hearing loss (HL) is the most prevalent sensorineural deficit, affecting about 20% of the global population (https://www.who.int/health-topics/hearing-loss, accessed on 24 May 2022), with nearly half due to genetic causes [1]. It is estimated that 1 in 500 newborns present with congenital HL, which profoundly affects language acquisition and consequently, social development [2]. Current standards of care for HL include hearing aids for people with diminished auditory function or cochlear implants, indicated for severe to profound HL. Unfortunately, there are no biological treatments available for HL and no interventions that restore auditory capacity to normal levels. During the last decade, many studies have investigated gene therapy strategies in animal models as treatments for genetic HL [3,4]. However, none of these experimental therapies have advanced to the clinic for treatment of humans. 

The auditory sensory epithelium is contained within the cochlea, the bony spiral structure of the inner ear, which can be accessed for therapeutic intervention via the round window membrane. Cochlear hair cells are organized in the organ of Corti within a tonotopic gradient: high frequencies are sensed at the base and low frequencies at the apex. The cochlea includes two types of hair cells: inner hair cells (IHCs), which convert mechanical signals into electrical signals and transmit them to contacting afferent neurons and on to the brain; and outer hair cells (OHCs), which serve as cochlear amplifiers, necessary for signal amplification and tuning.

To develop inner-ear gene therapy for genetic HL, our lab has focused on gene replacement for transmembrane channel-1 (*TMC1*) [5,6,7]. TMC1 forms the ion channel at the core of the mechanosensory transduction complex in auditory and vestibular hair cells [8,9]. Mutations in *TMC1* cause non-syndromic HL, which can be dominant (DFNA36) or recessive (DFNB7/11) [10,11]. Depending on the population, mutations in *TMC1* represent 3–8% of genetic HL [12]. Mice that lack *Tmc1* (*Tmc1^Δ/Δ^*) are profoundly deaf due to lack of sensory transduction and present with degeneration of cochlear hair cells beginning around four weeks of age [13]. 

In prior work from our lab, we used an AAV2/1 viral capsid to deliver mouse *Tmc1* cDNA sequence into *Tmc1^Δ/Δ^* mice and recovered their hearing capacity, albeit with high thresholds and only in the low frequency range [5]. The limited recovery was the result of viral transduction in IHCs, but not OHCs [5]. Even though the gene therapy strategy evoked a partial restoration of auditory thresholds in a limited hearing range, it demonstrated that hair cells can be targeted by AAV viruses and that the auditory function can be recovered in mice with profound genetic HL. In order to improve the extent of the recovery, non-conventional viral capsids were screened. Our lab discovered that the Anc80L065 capsid was able transduce both IHCs and OHCs in vivo [14]. Delivery of *Tmc1* using this capsid extended the level of auditory recovery at low and middle frequencies [6]. Recently, we showed that gene therapy for *Tmc1* using the synthetic AAV9-PHP.B capsid [15] further improved the recovery of auditory thresholds in *Tmc1^Δ/Δ^* mice [7]. Importantly, the AAV9-PHP.B capsid transduced IHCs and OHCs efficiently along the entire tonotopic gradient [16], which is required to enable broad-spectrum auditory recovery. 

While the prior work focused on gene therapy recovery in *Tmc1^Δ/Δ^* mice, there are no human *TMC1* mutations equivalent to that of *Tmc1^Δ/Δ^* mice. Over 65 human *TMC1* mutations have been described that cause DFNB7/11. Some mutations are associated with severe-to-profound congenital deafness and result from premature stop mutations, frame shift mutations, and other functional null mutations. In contrast, several point mutations that yield single amino acid substitutions cause milder phenotypes with moderate-to-severe hearing loss during the first decade of life [17]. For example, the human *TMC1* mutation p.N199I (c.596A > T) causes progressive moderate-to-severe hearing loss that develops during childhood [17]. Thus, we wondered whether the milder hearing loss phenotype in patients with p.N199I, and other similar mutations, may provide an expanded capacity for therapeutic intervention using a gene replacement strategy. 

We sought to study the effect of the missense p.N199I mutation by generating a humanized murine model harboring the homologous mutation in *Tmc1* (p.N193I) and developed a gene therapy strategy using gene replacement through AAV delivery. For this study, 143 mice were injected with one of three viral constructs at three different doses, and 117 mice were used as uninjected controls. We performed over 420 measurements of auditory function on the 260 mice in the study. We found that mice homozygous for the *Tmc1* p.N193I mutation have profound HL due to loss of mechanosensory transduction in cochlear hair cells. Mutant mice injected with AAV9-PHP.B carrying a functional copy of human *TMC1* (*hTMC1*) successfully recovered auditory capacity. Using a *CB6* promoter to drive the expression of *hTMC1* yielded auditory recovery that spanned the frequency spectrum. Additionally, the optimized gene therapy strategy efficiently recovered hearing capacity in *Tmc1^Δ/Δ^* mice, though to a lesser extent at high frequencies. We report unprecedented recovery of auditory thresholds using AAV9-PHP.B-optimized constructs encoding *hTMC1* in a humanized mouse model for genetic HL. These results demonstrate that early and efficient gene therapy intervention can restore auditory function in mice, suggesting potential translation for treatment of DFNB7/11 patients.

## 2. Materials and Methods

### 2.1. Mice

All procedures met NIH guidelines for the care and use of laboratory animals and were approved by the Institutional Animal Care and Use Committee (protocols #18-01-3610R and #20-02-4149R) at Boston Children’s Hospital (BCH).

We used wild-type C57BL/6J mice (Jackson Laboratories, Bar Harbor, ME, USA) with corrected *Ahl* allele, *Tmc1* mice with genotype *Tmc1^Δ/Δ^**; Tmc2^+/+^* on a C57BL/6J background [13], and knock-in mice with genotype *Tmc1^N193I/N193I^; Tmc2^+/+^* or *Tmc1^N193I/N193I^; Tmc2**^Δ/Δ^*on a C57BL/6J background with the *Ahl* allele corrected. Mouse line *Tmc1^N193I/N193I^; Tmc2^+/+^* was generated at the Mouse Gene Manipulation Core of the F.M. Kirby Neurobiology Center at BCH using the CRISPR/Cas9 system. The p.N193I (AAC > ATC) substitution in the *Tmc1* gene was introduced by disruption of the endogenous gene using SpCas9 (IDT, Coralville, Iowa, USA, Cat#181058) with a gRNA that specifically targeted the *Tmc1* sequence (5′-CAG GTGGATGTACGGCGTCAACATGG-3′) and homologous recombination using a template that spanned the whole exon (in upper case), part of the adjacent introns (in lower case) and contained the point mutation (in bold) (5′-ttggaagtcagggcttacCTCCGGTAACATGATGAGGCTGAAGGTCAACACAAAGAGAACCATGATGACGCCGTACATCCATCTGAGGAACAGGAAGTACGAGGCCACTGAGGAACCAAACTGACctgttacgaagaaaattaaaagactaaaggagagtcaaaacaacaacagcaccaatgaaaaac-3′).

The mutation was confirmed by Sanger sequencing analysis of a PCR product obtained from genomic DNA extracted from toe clips using lysis reagent (Viagen Biotech, Los Angeles, CA, USA) supplemented with Proteinase K (New England Biolabs, Ipswich, MA, USA) and genotyping primers (F: 5′-ATTGGGGTGGAAACCGATACGA-3′, R: 5′-TCTCCTGAGAGAAAGAAGAGCGA-3′). Amplification conditions were as follows: 98 °C for 30 s; 42 × (98 °C for 10 s; 66 °C for 30 s; 72 °C for 50 s); 72 °C for 2 min. PCR products were loaded into a 1.2% agarose gel stained with SYBR Safe (Invitrogen, Waltham, MA, USA) for electrophoresis. The amplicon band was identified in an iBright CL1500 Imaging System (Invitrogen) and separated from the rest of the gel for further purification using the Monarch DNA Gel Extraction Kit (New England Biolabs). 

Both males and females were used for experiments in approximately equal proportion. All mice were kept in a 12 h light/12 h dark cycle with accessibility to food and water *ad libitum*. 

### 2.2. Plasmids and AAV Production

Human TMC1 (NM_138691.3) codon-optimized sequence (*hTMC1*) was cloned into AAV2 plasmids downstream of a cytomegalovirus (CMV) or chicken beta-actin-derived (*CB6*) promoter and followed by a woodchuck hepatitis post-transcriptional regulatory element (*WPRE*), unless otherwise indicated. Vector was produced by triple transfection of suspension HEK293 cells grown in Expi293 media (Gibco, Waltham, MA, USA). Briefly, cells at ~2 × 10^6^ cells/mL were transfected with pALDX-80 helper plasmid, AAV2/AAV9PHP.B rep/cap plasmid, and *hTMC1* plasmid using PEIpro (Polyplus, Illkirch-Graffenstaden, France) formulated in OptiPRO SFM (Gibco). Seventy-two hours after transfection, cells were harvested by addition of Benzonase and 10× lysis buffer (500 mM Tris, 20 mM Magnesium Chloride, 10% Polysorbate 20, pH 8.0) followed by addition of 5 M Sodium Chloride to a final concentration of 500 mM. Lysates were clarified by centrifugation and stored at −80 °C until purification. 

Immediately prior to chromatography, lysates were filtered across a 0.45/0.22 µm polish filter (Millipore, St. Louis, MO, USA) and then passed over POROS CaptureSelect AAV9 (ThermoFisher, Waltham, MA, USA) affinity resin using an AKTA Pure instrument (Cytiva Life Sciences, Marlborough, MA, USA). Elution fractions with A280 values > 100 mAU were pooled and immediately neutralized with 10% volume of neutralization buffer (500 mM Bis-Tris Propane, 200 mM NaCl, 1% *w*/*v* Sucrose, 0.001% Poloxamer 188, pH 9.0). Following purification, the pooled affinity neutralized eluate was buffer-exchanged into final formulation buffer (dPBS, 5% *w*/*v* Sorbitol, 0.001% *w*/*v* Poloxamer 188, pH 7.0). Titer was determined using ddPCR as previously described [18]. The following titers of genome-containing particles were used: *AAV2/9-PHP.B-CMV-hTMC1-WPRE*: 3.26 × 10^13^ gc/mL or 6.3 × 10^13^ gc/mL; *AAV2/9-PHP.B-CB6-hTMC1 No WPRE*: 3.14 × 10^12^ gc/mL; *AAV2/9-PHP.B-CB6-hTMC1-WPRE*: 3.14 × 10^12^ gc/mL, 1.57 × 10^13^ gc/mL, 3.14–4.30 10^13^ gc/mL or 7.16 10^13^ gc/mL. Virus aliquots were stored at −80 °C.

### 2.3. Animal Surgery

All viral inner-ear injections were performed according the Institutional Animal Care and Use Committee at BCH protocols (#00001240 and #20-02-4149R) and were done via utricle injection as previously described [16]. In brief, P1-2 mice were anesthetized with hypothermia through 3 min of exposure to ice water. During the surgery (10–15 min), mice were kept on an ice pad. Using a stereo microscope (Stemi 2000, Zeiss, Oberkochen, Germany) for visualization, a small postauricular incision was made to expose the cochlea bulla and semicircular canals surrounding the utricle. After puncturing the temporal bone, a glass micropipette was inserted into the puncture to manually inject 1–1.2 µL of AAV at a constant rate. Only left ears were used for injections. Following the procedure, mice were placed on a heating pad until fully recovered, and standard postoperative care was applied

### 2.4. Electrophysiology 

Temporal bones were collected from P6–7 mice in Minimum Essential Media (Gibco). Overlying bone was removed with forceps and a needle until the cochlea could be accessed at the modiolus and separated from the vestibular portion of the inner ear. The organ of Corti was isolated from the modiolus and spiral ligament and anchored to a glass slide with a stainless-steel pin (Fine Science Tools, Foster City, CA, USA). After 1–2 days of incubation at 37 °C in minimum essential media supplemented with 10% fetal bovine serum (Life Technologies, Waltham, MA, USA) and 100 µg/mL ampicillin (Sigma-Aldrich, St. Louis, MO), the tectorial membrane was removed, and the tissue was placed under an Axioskop FS (Carl Zeiss) upright microscope and visualized using a 63× water-immersion objective. Patch clamp recordings in whole-cell configuration of inner hair cells from the basal portion of the cochlea were performed as previously described in Pan et al. (2013, 2018). Briefly, tissues were kept at room temperature and submerged in an artificial perilymph solution containing (in mM): 137 NaCl, 10 HEPES, 5.8 KCl, 5.6 d-glucose, 1.3 CaCl_2_, 0.9 MgCl_2_, 0.7 NaH_2_PO_4_, and vitamins (1:100) and amino acids (1:50) (ThermoFisher) at pH 7.4 and 311 mOsmol/kg. Recording electrodes (3–5 MΩ) were pulled from R6 capillary glass (King Precision Glass, Claremont, CA, USA) and filled with an internal solution containing (in mM): 140 CsCl, 5 HEPES, 5 EGTA-KOH, 3.5 MgCl_2_, 2.5 Na_2_ATP, and 0.1 CaCl_2_ at pH 7.4 and 284 mOsm/kg. Sensory transduction currents were recorded at a holding potential of −84 mV using an Axopatch 200B (Molecular Devices, San Jose, CA, USA). Currents were low-pass filtered at 2–5 kHz with a Bessel filter, sampled at 20 kHz with a 12-bit acquisition board (Digidata 1322A), recorded with pCLAMP software (Molecular Devices), and corrected for a 4 mV liquid junction potential. Hair cell bundles were deflected using a 4–5 µm tip diameter stiff glass probe mounted on a PICMA chip piezo actuator (Physik Instruments, Karlsruhe, Germany). The actuator was driven by an LPZT amplifier (Physik Instruments) and filtered with an 8-pole Bessel filter at 40 kHz to eliminate residual pipette resonance. Data were analyzed with Clampfit and Origin (OriginLab, Northampton, MA, USA).

### 2.5. FM1-43 Uptake

All experiments with membrane dye FM1-43 (Invitrogen) were performed at room temperature. FM1-43 stock solution (10 mM) was diluted to a working concentration (5 µM) in minimum essential media (Gibco). Mouse temporal bones were harvested at age P6, followed by microdissection of the organ of Corti. Freshly prepared FM1-43 was added to the culture for 10 s and then washed twice (5 min each wash) with minimum essential media. Live tissues were imaged with a 40× water immersion objective in an LSM 800 (Carl Zeiss) microscope.

### 2.6. Tissue Preparation and Confocal Immunofluorescence

Euthanization of 4- or 12-week-old mice was conducted via CO_2_ inhalation. Temporal bones were harvested, punctured at the round and oval windows and helicotrema, and fixed in 4% paraformaldehyde for 1 h at room temperature. Tissues were then decalcified in 120 mM EDTA for 16–24 h. Cochleas were sectioned into apical, middle, and basal portions. The organ of Corti was isolated and prepared for whole-mount processing by removal of the lateral wall, spiral limbus, and tectorial membrane. Tissues were permeabilized for 1 h in 0.25% Triton X-100, blocked for 1 h in 2.5% normal donkey serum, and stained at 4 °C overnight with rabbit anti-myosin 7a primary antibody (1:500 Proteus Biosciences, Ramona, CA, USA #25-6790). After washing with PBS, samples were incubated for 3–4 h with fluorophore-conjugated donkey anti-rabbit secondary antibody (1:400 Alexa Fluor 647: Thermo Fisher #A31573) and fluorophore-conjugated phalloidin (1:400 Alexa Fluor Plus 405: Thermo Fisher #A30104). Tissues were then mounted on a glass coverslip with Vectashield mounting medium (Vector Laboratories, Burlingame, CA, USA). Confocal imaging was performed using 10× air and 63× oil-immersion objectives with an LSM 800 (Carl Zeiss) microscope. Maximum intensity projection images were generated in ImageJ. Hair cell count analysis was performed manually.

### 2.7. Auditory Brainstem Response (ABR) Measurement

P16, 4-, 8-, or 12-week-old mice were anesthetized through intraperitoneal injection of a ketamine (100 mg/kg)/xylazine (15 mg/kg) cocktail. Mice were placed on a heating pad in a sound-proof chamber. The external auditory meatus was exposed through resection of the overlying skin and cartilage, and a custom speaker/microphone apparatus (EPL PXI Systems) was positioned directly above the entrance of the ear canal to deliver and record acoustic signals. Sound pressure at this location was calibrated for all stimulus frequencies before each test. Acoustic stimuli were delivered as 5 ms tone bursts at half-octave steps from 5.6 to 32 kHz and at 5 to 10 dB steps from 10 to 110 dB sound-pressure level (SPL) or until threshold was identified by the presence of a reproducible ABR waveform with a detectable peak 1. At each SPL, 512 responses to an alternating polarity stimulus were averaged. Electrical activity was recorded through subcutaneous needle electrodes behind the pinna (active electrode), at the vertex (reference), and at the hind leg (ground). ABR responses were amplified (10,000×), filtered (0.3 to 10 kHz), and digitized with LabVIEW software (Eaton-Peabody Laboratories). Electrode voltages and acoustic stimuli were sampled every 40 µs using a digital input-output board (National Instruments, Austin, TX, USA). Data were then plotted in Origin (OriginLab). 

### 2.8. Distortion Product Otoacoustic Emission (DPOAE) Measurement

DPOAE and ABR data were collected consecutively under identical conditions. DPOAE at 2f1-f2 were measured at half-octave f2 steps from 5.6 to 45.2 kHz and at 5 dB f2 steps from 10 to 80 dB SPL (f2/f1 = 1.2, L1–L2 = 10 dB). Threshold was determined after spectral averaging as the lowest L2 value to produce a DPOAE distinguishable from the surrounding noise floor and maintained at higher dB SPLs. Data were plotted in Origin (OriginLab).

### 2.9. RNA and DNA Isolation

Twelve-week-old animals were euthanized by cervical dislocation. The following organs were harvested, flash frozen immediately in liquid nitrogen and stored at −80 °C: injected (left) inner ear, contralateral (right) inner ear, brain and liver. RNA and DNA were isolated from frozen mouse cochlea using TRI-Reagent (Zymo Research, Irvine, CA, USA) and Direct-zol DNA/RNA Miniprep kits (Zymo Research). Briefly, frozen mouse cochleas were placed in a tube with a stainless-steel ball and TRI-Reagent. Samples were subjected to bead beating twice using a TissueLyser II (Qiagen, Hilden, Germany). Chloroform was added to samples and centrifuged to separate RNA, DNA, and protein. The top aqueous layer of RNA was carefully removed, followed by the interphase layer of DNA. Each layer was transferred to their respective tubes and diluted with 100% ethanol 1:1. RNA and DNA were subsequently purified through individual columns following the DNA/RNA Miniprep purification protocols. RNA and DNA from mouse brain and liver were purified using the All-Prep DNA/RNA kit (Qiagen) according to the manufacturer’s protocol. Peripheral tissues were lysed similar to cochleas using the kits lysis buffer instead of TRI-Reagent.

### 2.10. Quantitative PCR

RNA and DNA were quantified according to the manufacturer’s protocols using the Quant-iT RNA Assay Kit (ThermoFisher) and the Quant-iT dsDNA Assay Kit (ThermoFisher) respectively; 100 ng of RNA underwent cDNA synthesis using the RT^2^ Easy First Strand Kit (Qiagen) according to the manufacturer’s protocol. 

To assess vector genomes and transgene expression, qPCR was performed on gDNA and cDNA using PrimeTime Gene Expression 2× Master Mix (IDT), a fluorescently labeled 5′ nuclease probe (IDT), and hTMC1 opt primers with the following sequences: forward primer, 5′-GATCAGGATGGTCACGTATG-3′ and reverse primer 5′-GCTGGG AGACAATGGTAG-3′ and CFX-96 (BioRad, Hercules, CA, USA) instruments. A standard curve was prepared using a gBlock standard established over a six-fold dilution series from 5E1–5E6 copies per reaction. Samples were analyzed by the absolute quantification method against the standard curve using CFX Maestro software (BioRad). Transgene expression was normalized and represented as copies/1 µg RNA. Vector genome copies were normalized to amount and reported as copies/1 µg DNA. 

### 2.11. Statistical Analyses

Normality tests were performed on the data prior to running statistical analysis. All figures present datasets that do not show normal distribution, and therefore, non-parametric tests were performed. For comparing two different groups (i.e., electrophysiology recording), Mann–Whitney test was used. For comparing datasets with more than two groups (i.e., hair cell counts), Kruskal-Wallis followed by Dunn test for multiple comparisons was applied. A *p* value equal to or less than 0.05 was considered significant. Statistical analysis was performed using Origin (OriginLab). Individual traces and mean values ± S.E.M. are presented in all figures. All values and statistics are presented in Appendix A. 

## 3. Results

### 3.1. Characterization of p.N193I Mice

Patients homozygous for the p.N199I mutation in TMC1 present with moderate hearing loss, which progresses during the first decade of life [17]. We hypothesized that since this recessive mutation causes moderate but not profound HL, tissue degeneration may occur at later stages, suggesting the potential for an expanded capacity for therapeutic intervention. To generate a humanized mouse model for HL, we used CRISPR-Cas9 technology coupled with homologous recombination to specifically introduce an A > C substitution, which converted an AAC codon to ATC codon in the mouse genomic DNA (Figure 1A). The DNA substitution in the mouse *Tmc1* gene yielded a p.N193I point mutation, homologous to the p.N199I mutation in humans. The N193 site occurs within the predicted first transmembrane domain of TMC1 [9,19] and is highly conserved among vertebrate clades (Figure 1B). As such, we hypothesized the residue is critical for TMC1 function and normal hearing in mice and humans. We tested the hearing capacity of the *Tmc1^N193I/+^* and *Tmc1^N193I/N193I^* mice at postnatal day (P) 28 by recording auditory brainstem responses (ABRs) at a range of frequencies (5.6 to 32 kHz). We found that the *Tmc1^N193I/+^* mice had normal hearing, with an average ABR threshold of 17.2 ± 2.2 decibels (dB) at 16 kHz (Figure 1C,D). We also recorded distortion products otoacoustic emissions (DPOAEs), an assay for outer hair cell function, and found that *Tmc1^N193I/+^* mice had normal DPOAE thresholds compared to *Tmc1^+/+^* mice (Figure 1D). On the contrary, *Tmc1^N193I/N193I^* mice were deaf, evident from the absence of ABR (up to 110 dB) and DPOAEs (up to 80 dB) thresholds at 1 month of age (Figure 1C,D). We tested the hearing capacity of *Tmc1^N193I/N193I^* mice at P16, which is close to the hearing onset (~P12), to determine if these animals were born deaf or if the HL progressed rapidly after hearing onset. We found that whereas the *Tmc1^N193I/+^* mice had normal hearing thresholds, P16 *Tmc1^N193I/N193I^* mice had no ABR or DPOAE thresholds (Figure 1D). Therefore, unlike humans with the p.N199I mutation who have moderate, progressive HL, mice carrying the p.N193I mutation are profoundly deaf from birth. 

To delve into the mechanism underlying the HL phenotype, we evaluated hair cell sensory transduction properties by examining uptake of the vital dye FM1-43, a large organic compound that can permeate functional mechanosensory transduction channels [20,21,22]. We determined that IHCs and OHCs from *Tmc1^N193I/N193I^; Tmc2^Δ/Δ^* mice had no FM1-43 fluorescence (Figure 1F), indicating a lack of sensory transduction in homozygous mice. We also measured sensory transduction in mutant mice by recording currents elicited by direct bundle displacement from basal IHCs. Cells from *Tmc1^N193I/N193I^; Tmc2^Δ/Δ^* mice had no transduction currents (−0.8 ± 0.7 pA, *n* = 12; Figure 1E and Appendix A), whereas cells from *Tmc1^N193I/+^; Tmc2^Δ/Δ^* mice had large, stimulus-induced transduction currents (−211.2 ± 16.4 pA, *n* = 13, *p* value < 0.0001, Mann–Whitney test; Figure 1E and Appendix A), indicating that the deafness phenotype of mutant mice is caused by the loss of sensory transduction in auditory hair cells, similar to that of IHCs from *Tmc1^Δ/Δ^* mice [8]. 

Next, we compared the number of surviving hair cells from P28 *Tmc1^N193I/+^*, *Tmc1^N193I/N193I^* and *Tmc1^Δ/Δ^* mice. Heterozygous mice showed no signs of hair cell loss in the different regions of the cochlea (Figure 1G,H and Appendix A). On the contrary, *Tmc1^N193I/N193I^* mice showed a reduced number of OHC at the cochlea middle and base compared to heterozygous animals (* *p* = 0.0388 for the middle and * *p* = 0.0475 for the base *Tmc1^N193I/+^* (*n* = 5) vs *Tmc1^N193I/N193I^* (*n* = 9), Kruskal−-Wallis followed by Dunn’s test; Figure 1G,H and Appendix A). Also, P28 *Tmc1**^Δ/^**^Δ^* mice showed significantly fewer OHCs at the middle and base compared to the phenotypically normal *Tmc1^N193I/+^* mice (***p* = 0.0012 for the middle and *** *p* = 0.0007 for the base *Tmc1^N193I/+^*, *n* = 5, vs *Tmc1**^Δ/^**^Δ^*, *n* = 4, Kruskal−-Wallis followed by Dunn’s test; Figure 1G,H and Appendix A). Interestingly, *Tmc1^N193I/N193I^* mice had, on average, more surviving OHC than *Tmc1**^Δ/^**^Δ^* mice, although the difference was not statistically significant (Figure 1G,H and Appendix A). This finding suggests that even though both murine models present with profound deafness due to loss of sensory transduction, tissue degeneration was delayed in the humanized *Tmc1^N193I/N193I^* line.

### 3.2. Hearing Recovery Using PHP.B-CMV-hTMC1

To optimize a gene therapy approach suitable to treat human patients with DFNB7/11, we engineered viral vectors encoding a strong cytomegalovirus (*CMV*) promoter, followed by a codon-optimized version of human *TMC1* (*hTMC1*) and a woodchuck hepatitis virus post-transcriptional regulatory element (*WPRE*) sequence (for short: *CMV-hTMC1*), and packaged it into AAV9-PHP.B viral capsids. We injected ~1 µL (3 × 10^10^ gc), a dose that has been shown to be effective in transducing ~100% of IHCs and 95% of OHCs (Lee et al., 2020), into the left utricle of *Tmc1**^Δ/^**^Δ^* mice at P1-2, and recorded ABRs and DPOAEs after 4 weeks. Because the success rate of the surgical procedure is variable and we do not have a method to distinguish poor-performing constructs from failed injections, we recorded ABRs and DPOAEs from all surviving animals injected with the various viral constructs and plotted their hearing performance as a function of frequency. *Tmc1**^Δ/^**^Δ^* animals injected with *CMV-hTMC1* showed prominent recovery of auditory thresholds at low and middle frequencies (Figure 2A). We wondered whether doubling the viral dose would yield an improvement in the recovery and injected P1-2 *Tmc1**^Δ/^**^Δ^* mice with 6 × 10^10^ gc. Increasing the viral dose clearly improved the recovery at 22.6 kHz (Figure 2B). The improvement in the auditory capacity was concomitant with a recovery of DPOAEs (Figure 2E,F), underscoring the necessity for OHC amplification to achieve normal hearing levels. However, increasing the titer of the *CMV-hTMC1* did not improve the recovery at 32 kHz, the highest frequency assessed (Figure 2A,E). Importantly, we showed for the first time that *hTMC1* is effective for recovering auditory function in *Tmc1**^Δ/^**^Δ^* mice in a manner similar to the mouse *Tmc1* sequence [7]. 

Since the *CMV-hTMC1* proved efficacious in *Tmc1**^Δ/^**^Δ^* mice at low and middle frequencies, we tested the construct in the humanized *Tmc1^N193I/N193I^* model. Knock-in mice injected with *CMV-hTMC1* (3 × 10^10^ gc) showed clear recovery of ABR thresholds at low and middle frequencies, up to 22.6 kHz (Figure 2C,G).

Interestingly, increasing the dose to 6 × 10^10^ gc showed an improvement in the recovery at 32 kHz frequency, with a 45 dB threshold as the best recorded performance (Figure 2D,H). These results demonstrate that the *CMV-hTMC1* construct is effective for recovering auditory capacity in different mouse lines, especially in the humanized *Tmc1^N193I/N193I^* model, which not only fully recovered auditory low and middle frequencies but also showed a modest recovery of hearing thresholds at high frequencies (Figure 2A–H).

Next, we evaluated whether the recovery was durable by re-testing the same *Tmc1**^Δ/^**^Δ^* and *Tmc1^N193I/N193I^* treated animals at 8- and 12-weeks of age (Figure 2I–P). *Tmc1**^Δ/^**^Δ^* mice injected with the 3 × 10^10^ gc dose showed an elevation of hearing thresholds over time (Figure 2I,M). For *Tmc1**^Δ/^**^Δ^* mice injected with 6 × 10^10^ gc, hearing thresholds at later time points were preserved, although with a slight elevation at 12 weeks compared to 4 weeks, especially at high frequencies (Figure 2J,N). Further, 8- and 12-week-old ABR and DPOAE thresholds of *Tmc1^N193I/N193I^* injected mice were more robust, even for those injected with the lower dose (Figure 2K,L,O,P). All mean ± S.E.M. and *n* values from Figure 2 are shown in Appendix A.

### 3.3. Hearing Recovery Using PHP.B-CB6-hTMC1 (+/− WPRE)

Hearing loss is more severe at high frequencies in both human and mouse, and recovery of high frequency auditory capacity by gene therapy has not been successful [23]. We were encouraged by the results showing recovery of ABR and DPOAE thresholds for the *Tmc1^N193I/N193I^* line at 32 kHz (Figure 2D,H) and decided to explore alternative promoters to boost recovery and durability. To that end, we generated a construct encoding a modified chicken beta-actin-derived promoter (termed *CB6*) [24], which is not as strong as the *CMV* promoter but may provide consistent, constitutive activity with less cell-to-cell variability (data not shown). We cloned the *CB6* promoter upstream of the *hTMC1* and the *WPRE* sequences (for short: *CB6-hTMC1 + WPRE*) and packaged the construct into AAV9-PHP.B viral capsids. The vectors were injected into P1-2 *Tmc1**^Δ/^**^Δ^* and *Tmc1**^N193I/^**^N193I^* mice. *Tmc1**^Δ/^**^Δ^* mice treated with the *CB6-hTMC1 + WPRE* (3 × 10^10^ gc) construct showed ABR and DPOAE recovery at lower and middle frequencies (Figure 3A, E). Importantly, around 50% of the injected animals exhibit robust ABR thresholds, with recovery at frequencies up to 22.6 kHz (best performance of 20 dB ABR threshold; Figure 3A), proving that the *CB6* promoter effectively recovered hearing capacity in *Tmc1**^Δ/^**^Δ^* mice. We then tested the *CB6-hTMC1 + WPRE* construct in the humanized *Tmc1**^N193I/^**^N193I^* line and discovered that 10 of 13 treated mice presented full recovery of ABRs and DPOAEs with thresholds equivalent to those of wild-type mice across the entire range of frequencies tested (Figure 3C,G). This is the first report of complete hearing recovery that spans the frequency spectrum for animals that are otherwise completely deaf.

Next, we tested whether the inclusion of the *WPRE* element was important for the extent and durability of hearing recovery. We removed the *WPRE* element from the viral construct to yield *CB6-hTMC1-No WPRE* vectors. Similar to the *Tmc1**^Δ/^**^Δ^* mice treated with *CB6-hTMC1 + WPRE*, *CB6-hTMC1-No WPRE* injected animals recovered ABR thresholds from 5.6 to 22.6 kHz (Figure 3B,F). The same trend was observed for the *CB6-hTMC1-No WPRE* in the *Tmc1**^N193I/^**^N193I^* treated animals, in which five of six mice presented ABR and DPOAE thresholds that completely overlapped the *Tmc1**^N193I/^**^+^* recordings for all the frequencies assessed (Figure 3D,H). Interestingly, whereas the auditory recovery was durable for up to 12 weeks post-injection for the *Tmc1**^Δ/^**^Δ^* and *Tmc1^N193I/N193I^* mice treated with *CB6-hTMC1 + WPRE* (Figure 3I,K,M,O), the hearing thresholds of animals injected with the *CB6-hTMC1-No WPRE* construct increased at later time points (Figure 3J,L,N,P), especially for the *Tmc1**^Δ/^**^Δ^* mice, which exhibit almost no detectable recovery at 8- and 12-weeks of age (Figure 3J,N). These results indicate that the *WPRE* element is important for long-term durability of hearing recovery in both HL mouse models. All mean ± S.E.M. and *n* values from Figure 3 are shown in Appendix A.

### 3.4. PHP.B-CB6-hTMC1 Optimal Dose

After identifying the most effective construct as *CB6-hTMC1 + WPRE*, we next tested the optimal viral dose. We injected *Tmc1**^N193I/^**^N193I^* and *Tmc1**^Δ^^/^**^Δ^* mice at P1-2 with doses that ranged from 3 × 10^9^ gc to 7 × 10^10^ gc and evaluated their hearing capacity after 4 weeks. We determined the pure tone average (PTA) for each animal by calculating the average ABR or DPOAE threshold recorded for each pure tone and plotted them as a function of viral dose (Figure 4). A Boltzmann equation was used to fit the data and obtain a half maximal dose (EC_50_). Untreated controls are included in the plot for comparison (Figure 4). We determined the ABR EC_50_ values for *Tmc1**^Δ/^**^Δ^* and *Tmc1**^N193I/^**^N193I^* mice to be 2.5 × 10^10^ gc and 1.5 × 10^10^ gc, respectively (Figure 4A,B). Additionally, *Tmc1**^N193I/^**^N193I^* treated with the middle (3 × 10^10^ gc) and highest dose (7 × 10^10^ gc) present ABR PTAs similar to *Tmc1**^N193I/^**^+^* mice (Figure 4B), which have normal hearing (Figure 1C,D). Importantly, the minimum dose that yielded best performance was 3 ×10^10^ gc since the lowest PTA in treated mutant mice were detected in animals injected with this dose (Figure 4). All mean ± S.E.M. and *n* values from Figure 4 are shown in Appendix A. 

To evaluate potential toxicity of the gene therapy construct, we injected wild-type mice with *CB6-hTMC1 + WPRE* (3 × 10^10^ gc) and recorded ABRs and DPOAEs at 4 weeks of age. Results showed that eight of nine injected mice had normal hearing at P28 (Appendix A). One mouse had elevated ABR and DPOAE thresholds and presented with tissue covering the external ear canal, which is indicative of conductive HL and was probably a consequence of the surgical intervention. Injected wild-type mice did not show elevation of ABR or DPOAE thresholds at 8- and 12-weeks of age (Appendix A), suggesting that the therapeutic construct had minimal toxicity in normal hearing mice. Importantly, wild-type injected cochleas did not show hair cell degeneration, since the number of surviving hair cells at 12 weeks of age was similar to their uninjected counterparts (Appendix A). 

We also performed biodistribution analysis on *Tmc1**^Δ/^**^Δ^*- and *Tmc1**^N193I/N193I^*-injected animals to detect viral DNA and RNA in the injected (left) ear and off-target sites: the contralateral (right) ear, brain and liver. As expected, we found viral DNA and RNA in the injected ear and in the contralateral ear (Appendix A), consistent with previous results showing viral distribution in contralateral ears [14,25,26]. We also found high quantities of viral DNA and RNA in the brain and at lower levels in the liver of injected animals (Appendix A). Importantly, although viral DNA and RNA was detected in various tissues, we did not observe any overt consequences, suggesting minimal toxicity of the AAV9-PHP.B capsid and the *CB6-hTMC1 + WPRE* construct.

### 3.5. Hair Cell Survival

To evaluate whether these gene therapy strategies promoted hair cell survival, cochlear tissues from treated and control mice were collected at 12-weeks of age. Cochleae were fixed and stained against myosin 7a, which specifically labels IHCs and OHCs, to assess hair cell survival. As previously determined [7], uninjected *Tmc1**^Δ/^**^Δ^* mice showed prominent degeneration of the organ of Corti at 12 weeks of age, with only a few surviving hair cells at the apical region and complete loss of OHCs at the middle and base (average IHC/100 µm: 6.9—apex, 4.2—middle, 2.8—base; average OHC/100 µm: 11.5—apex, 0—middle, 0—base; Figure 5A,B and Appendix A). To capture the full range of responses, we selected mice for histological analysis that had good ABR recovery and mice with poor recovery. As previously reported (Nist-Lund et al., 2019; Wu et al., 2021), we found that mice with poor ABR recovery also had poor hair cell survival. *Tmc1**^Δ/^**^Δ^* mice injected with either titer of the *CMV-hTMC1* constructs that had good ABR thresholds also showed survival of both IHCs and OHCs; however, only those treated with 6 × 10^10^ gc dose had surviving OHCs at the base (Figure 5A,B and Appendix A). The same trend was observed with *Tmc1**^Δ/^**^Δ^* mice treated with the *CB6-hTMC1 + WPRE* construct, where the animals injected with 7 × 10^10^ gc had higher hair cell counts at the base compared to those treated with 3 × 10^10^ gc (Figure 5A,B and Appendix A).

For the 12-week-old uninjected *Tmc1**^N193I/N193I^* mice, substantial loss of both IHCs and OHCs for all the tonotopic regions of the cochlea was evident (average IHC/100 µm: 7.9—apex, 4.6—middle, 4.3—base; average OHC/100 µm: 21.2—apex, 0.1—middle, 0.1—base; Figure 5A,B). Delivery of the *CMV-hTMC1* construct at both titers promoted prominent hair cell survival in *Tmc1**^N193I/N193I^* mice (Figure 5C,D and Appendix A). *Tmc1**^N193I/N193I^* mice treated with the *CB6-hTMC1 + WPRE* construct had higher IHC and OHC survival in every tonotopic region, even at the lower dose, 3 × 10^10^ gc (Figure 5C,D and Appendix A). Survival of IHCs and OHCs at the basal region is necessary for high frequency auditory sensitivity; therefore, it is relevant that our gene therapy strategy prevented hair cells degeneration across all cochlear turns. Importantly, the increase in the hair cell counts for the *CB6-hTMC1 + WPRE* injected *Tmc1**^N193I/N193I^* animals was achieved using a dose of 3 × 10^10^ gc. 

## 4. Discussion

In the recent years, significant improvements in gene therapy as a treatment for HL have been made in animal models [3,4,23]. The discovery of viral capsids that target hair cells with higher efficiency has enabled specific delivery of genes involved in sound perception. However, further optimization of viral constructs in animal models of genetic HL is needed prior to translation to the clinic. 

Due to a lack of appropriate models, gene therapy for recessive DFNB7/11 HL has not been tested in a mouse model that bears a human *TMC1* mutation. Prior work in *Tmc1^Δ/Δ^* mice provided important proof-of-concept data [5,6,7], but as there are no equivalent mutations in human *TMC1*, questions remain regarding relevance to the human condition. As such, we sought to develop a mouse model bearing a recessive human *TMC1* mutation and focused on the p.N199I mutation [17]. We reasoned that since patients carrying this mutation have a milder form of HL, there may be an expanded capacity for clinical intervention. Patients homozygous for the p.N199I mutation present with moderate HL at 3–5 years of age, which progresses over the next decade to more severe HL. Given the phenotype and the role of TMC1 as the hair cell sensory transduction channel, we hypothesized that the p.N199I mutation may yield a hypofunctional ion channel which may be sufficient to allow for some auditory function and survival of auditory hair cells during the first decade of life. However, to our surprise, when we generated and validated the mouse model bearing the equivalent mutation, p.N193I, we discovered that the homozygous mice were profoundly deaf at the earliest time point tested, P16. Furthermore, the mice lacked sensory transduction entirely, suggesting that, contrary to our hypothesis, the p.N193I version of TMC1 was not hypofunctional; rather, the single amino acid substitution caused a complete loss of function. These unanticipated results raise several important and interesting questions. 

Given that TMC1 has the same function in mice and humans and both sequences are highly conserved [27], we expect that the effect of the mutation on protein function is the same for both species. Therefore, if the p.N199I mutation causes profound loss of TMC1 function, how is it that humans bearing this mutation retain some residual auditory function? We reasoned that a compensatory mechanism that overcomes the loss of TMC1 function must exist in humans during the first decade of life. While the identity of this compensatory mechanism remains unknown, we suspect TMC2 may be playing an expanded or compensatory role in human cochlear hair cells. In mouse cochlear hair cells, TMC2 expression rises during the first postnatal week and is sufficient to support sensory transduction in inner and outer hair cells [8,13]. However, TMC2 expression begins to fall at the end of the first postnatal week while TMC1 expression begins to rise [13]. While the temporal expression patterns of TMC1 and TMC2 in human cochlear hair cells is unknown, perhaps heterochronic differences between mouse and human cochlear development may result in a longer or compensatory expression profile for human TMC2. If so, the residual auditory function in humans with p.N199I mutations may be a consequence of TMC2 function. 

Regardless of the underlying cause of the residual hearing function in patients with p.N199I mutations, the presence of any auditory function implies there must be some surviving and functional hair cells during the first decade of life. As such, we suspect that these hair cells may be viable targets for gene therapy intervention and may provide an expanded capacity for auditory recovery. To investigate this hypothesis, we tested three viral constructs in *Tmc1^N193I/N193I^* mice and in the *Tmc1^Δ/Δ^* model. First, we were interested to determine if a codon optimized version of human TMC1 was capable of restoring auditory function in mice. Indeed, in both lines, we observed substantial recovery of ABR and DPOAE thresholds and enhanced survival of auditory hair cells, confirming that the codon-optimized human coding sequence could substitute for the mouse sequence. Next, we investigated a modified chicken beta-actin-promoter, *CB6*, which is known to drive stable expression in a variety of cell types [24]. In addition, we investigated the necessity of the post-transcriptional regulatory element, *WPRE*, for the durability of auditory recovery. We found that together, the *CB6* promoter and the *WPRE* element enhanced the auditory recovery and promoted the durability of recovery in both animal models. Importantly, there was a remarkable level of auditory recovery in the *Tmc1^N193I/N193I^* mice, with ABR and DPOAE thresholds that were indistinguishable from wild-type thresholds across the entire auditory spectrum for 10 of 13 mice injected with the AAV9-PHP.B*-CB6-hTMC1-WPRE* construct. Furthermore, the recovery was durable up to 12 weeks post-injection. Injected *Tmc1^Δ/Δ^* mice also had remarkable and durable recovery with the same viral construct at a higher dose, but the recovery was reduced at the high frequency end of the spectrum. We did notice a decline in auditory function in both p.N193I mice and *Tmc1^Δ/Δ^* mice injected with the construct lacking the *WPRE* element, underscoring the necessity of the *WPRE* sequence for durable recovery of function. When the same viral construct, AAV9-PHP.B*-CB6-hTMC1-WPRE,* was injected into wild-type mice, we found no loss of auditory function in eight of nine mice. In the one mouse with elevated thresholds, we suspect the injection procedure itself may have damaged the middle ear structures, as histological analysis revealed a full complement of surviving hair cells in the injected ear. 

Due to age-related decline in viral tropism for mouse outer hair cells (Lee et al., 2020), we did not explore viral injection beyond the first postnatal week. However, we note that previous work in adult non-human primates showed that the AAV9-PHP.B viral capsid, with a CBA promoter and a WPRE sequence, was capable of transfecting and driving the expression of a reporter transgene in IHCs and OHCs [28]. Interestingly, the viral titers that were effective in targeting hair cells along the different tonotopic regions of the primate cochlea were similar to those found to be effective for mice in this study (3.5 and 7 × 10^11^ dose, injecting 10 µL) [28]. Recently, van Beelen et al. [29] showed that AAV9-PHP.B viral capsids carrying a CMV promoter and the GFP coding sequence can transduce and drive expression of GFP in fetal and adult human hair cells in *ex vivo* explants of biopsied inner-ear tissue. These results suggest that the AAV9-PHP.B-*CB6-hTMC1 + WPRE* could efficiently target hair cells and drive expression of *hTMC1* in non-human primate hair cells, and perhaps ultimately in humans as well. 

In our mouse studies, biodistribution analysis revealed viral DNA and RNA in the contralateral ear, brain and liver, suggesting that in neonatal mice, some vector does escape the injected ear. However, as no overt consequences were noted in these animals, we conclude that systemic biodistribution of the viral construct does not cause noticeable toxicity in mice. 

The difference in the response to viral gene therapy between the *Tmc1^N193I/N193I^* mice and *Tmc1^Δ/Δ^* models raises important questions. Why did the *Tmc1^N193I/N193I^* mice respond with complete recovery of auditory function, whereas the knock-out model did not? Since the knock-out model lacks expression of the full-length TMC1 protein, we suspect the consequences of this mutation may be more severe, leading to collapse of the hair cell transduction apparatus, more rapid hair cell degeneration, and somewhat diminished ability to respond to gene therapy intervention. On the other hand, the *Tmc1^N193I/N193I^* mice generate a full-length, albeit non-functional, TMC1 protein. The presence of the protein, even if non-functional, may provide structural support for the transduction complex and help stabilize the apparatus. Alternatively, the mutation may indirectly disrupt the ion channel pore but may leave other TMC1 functions intact. Since the location of N193 is in the first transmembrane domain, the Alpha-fold structure predicts that this residue is within range to interact with Q581 in the seventh transmembrane domain, which lines the channel pore [9,19]. Since the polar side chains are facing each other at 2.8–3.4 Å, mutation at the N193 site may destabilize the pore region but leave other regions of TMC1 unaffected.

Although the *Tmc1^N193I/N193I^* mice did not mimic the milder and progressive auditory phenotype seen in humans with p.N199I mutations, the remarkable gene therapy recovery in the p.N193I model may suggest that the recovery expected in the presence of the milder human phenotype may be equal to or greater than the recovery documented for mice. Furthermore, we wonder whether this level of recovery may be representative not only of the response expected for p.N199I patients, but perhaps more broadly for all recessive *TMC1* point mutations that yield a full-length but non-functional TMC1 protein. More than half of recessive mutations in human TMC1 (40/70) are single amino acid substitutions, raising the possibility that patients harboring these mutations may have an expanded capacity to benefit from TMC1 gene therapy. 

The *Tmc1^Δ/Δ^* mice also had remarkable auditory recovery in the low-frequency range. We suggest these mice may be a better model of functional null mutations in human *TMC1* such as frame shifts, premature stop codons, and so forth. In conclusion, we suggest that the AAV9-PHP.B*-CB6-hTMC1-WPRE* vector may be well-suited as a gene therapy reagent for treatment of patients with all forms of recessive *TMC1* hearing loss.

## Figures and Tables

**Figure 1 biomolecules-12-00914-f001:**
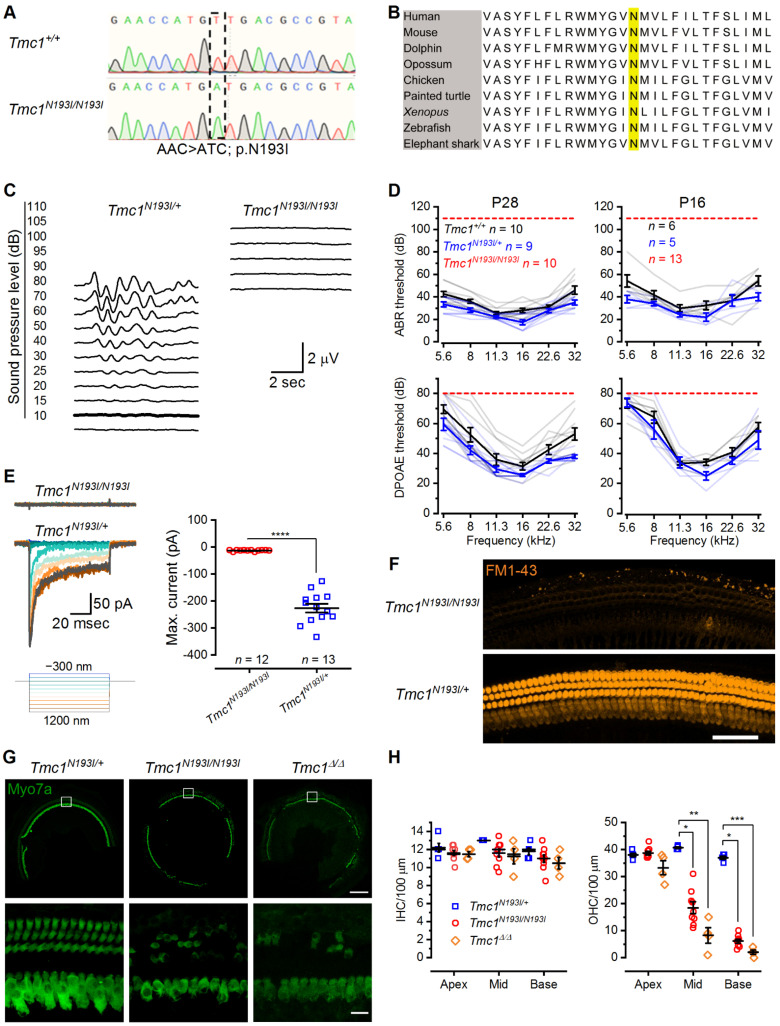
p.N193I mouse line generation and characterization. (**A**) Chromatogram of Sanger sequencing showing the wild-type (AAC) and the p.N193I (ATC) mouse genomic DNA sequences. (**B**) Multiple alignment of TMC1 first transmembrane domain containing the highly conserved N (yellow). Human: *Homo sapiens*; Mouse: *Mus musculus*; Dolphin: *Tursiops truncatus*; Opossum: *Monodelphis domestica*; Chicken: *Gallus gallus*; Painted turtle: *Chrysemys picta bellii*; *Xenopus*: *Xenopus tropicalis*; Zebrafish: *Danio rerio*; Elephant shark: *Callorhinchus milii*. (**C**) Representative ABR waveforms from *Tmc1^N193I/+^* and *Tmc1^N193I/N193I^* mice recorded at 4 weeks of age for 16 kHz tone bursts at increasing sound-pressure levels (dB). Thresholds are indicated by thicker traces and were determined by the presence of peak 1. Scale bar applies to both recordings. (**D**) ABR (up) and DPOAE (down) thresholds (dB) as a function of the frequency (kHz) for P28 (left) and P16 (right) wild-type (black), *Tmc1^N193I/+^* (blue) and *Tmc1^N193I/N193I^* (red) mice. Mean ± S.E.M. thresholds are shown in darker tones, and lighter traces correspond to individual recordings. (**E**) Mechanosensory transduction currents recorded from IHCs of *Tmc1^N193I/+^*; *Tmc2^Δ/Δ^* (blue) and *Tmc1^N193I/N193I^*; *Tmc2^Δ/Δ^* (red) P7-8 mice. Representative currents recorded at −84 mV and stimulation protocol (left). Mean ± S.E.M. maximal current amplitudes recorded for 12 (*Tmc1^N193I/N193I^*; *Tmc2^Δ/Δ^*) or 13 (*Tmc1^N193I/+^*; *Tmc2^Δ/Δ^*) individual cells, from 3–5 different animals (right). **** *p* < 0.0001, Mann–Whitney test. (**F**) Confocal microscopy images of live cochleas focused at the hair cell level, from *Tmc1^N193I/+^*; *Tmc2^Δ/Δ^* and *Tmc1^N193I/N193I^*; *Tmc2^Δ/Δ^* P6 mice incubated with 5 μM FM1-43 (orange) for 10 s. Scale bar: 50 μm. (**G**) Representative confocal images from middle cochlear sections of *Tmc1^N193I/+^*, *Tmc1^N193I/N193I^* and *Tmc1^Δ/Δ^* P28 mice immunostained against myosin 7a (green). Scale bars: top panels (10×) 200 µm; lower panels (63×) 15 µm. White boxes (above) indicate region shown at higher magnification (below). (**H**) Mean ± S.E.M. hair cell counts per 100 µm sections of IHC (left) and OHC (right) from P28 *Tmc1^N193I/+^* (*n* = 5), *Tmc1^N193I/N193I^* (*n* = 9) and *Tmc1^Δ/Δ^* (*n* = 4) measured in the apex, middle and base cochlear turns. Individual samples are included in the scatterplots * *p* < 0.05; ** *p* < 0.01; *** *p* < 0.001; Kruskal—Wallis followed by Dunn’s multiple comparison test.

**Figure 2 biomolecules-12-00914-f002:**
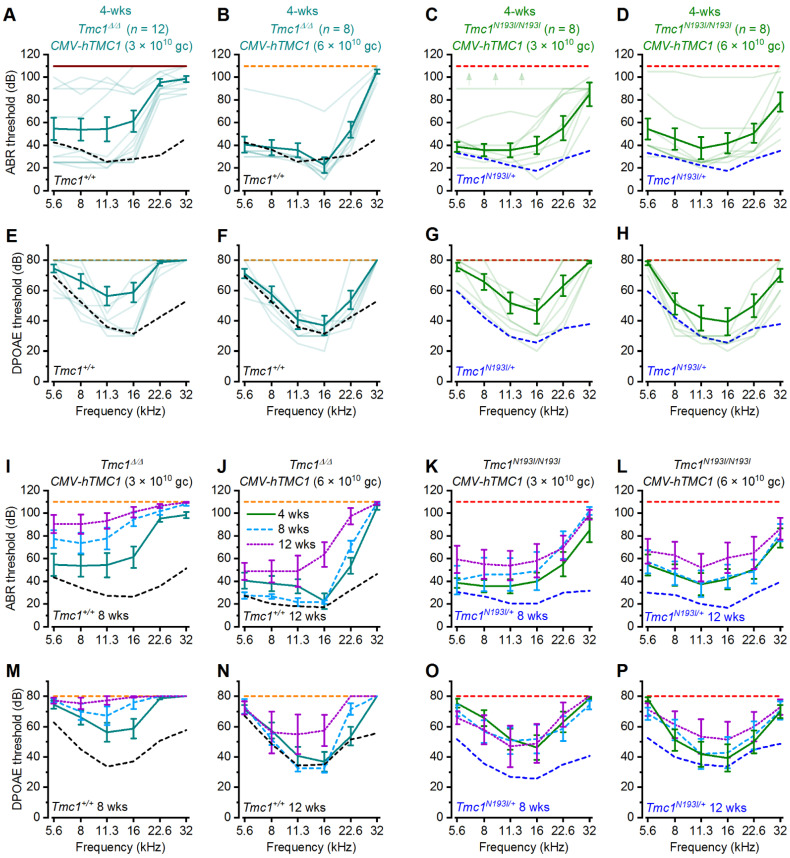
Deaf mice treated with AAV2/9-PHP.B*-CMV-hTMC1* efficiently recover low and middle frequency auditory thresholds. ABR (**A**–**D**) or DPOAE (**E**–**H**) thresholds as a function of frequency of *Tmc1^Δ/Δ^* (**A**,**B**), (**E**,**F**) and *Tmc1^N193I/N193I^* (**C**,**D**), (**G**,**H**) mice injected at P1-2 with *CMV-hTMC1* at a titer of 3 × 10^13^ gc/mL (**A**,**C**,**E**,**G**) or 6 × 10^13^ gc/mL (**B**,**D**,**F**,**H**) at 4 weeks of age. Mean ± S.E.M. thresholds shown in bold traces and individual recording are shown as lighter traces. Mean ± S.E.M. ABR (**I**–**L**) or DPOAE (**M**–**P**) thresholds against frequency across time for injected 4-, 8-, and 12-week-old mice. Dashed lines indicate mean ABR or DPOAE thresholds for uninjected control mice: *Tmc1^Δ/Δ^* (orange), *Tmc1^+/+^* (black) (4-week-old: (**A**,**B**,**E**,**F**); 8-week-old: (**I**,**M**); 12-week-old: (**J**,**N**) and *Tmc1^N193I/N193I^* (red), *Tmc1^N193I/+^* (blue) (4-week-old: (**C**,**D**,**G**,**H**); 8-week-old: (**K**,**O**); 12-week old: (**L**,**P**).

**Figure 3 biomolecules-12-00914-f003:**
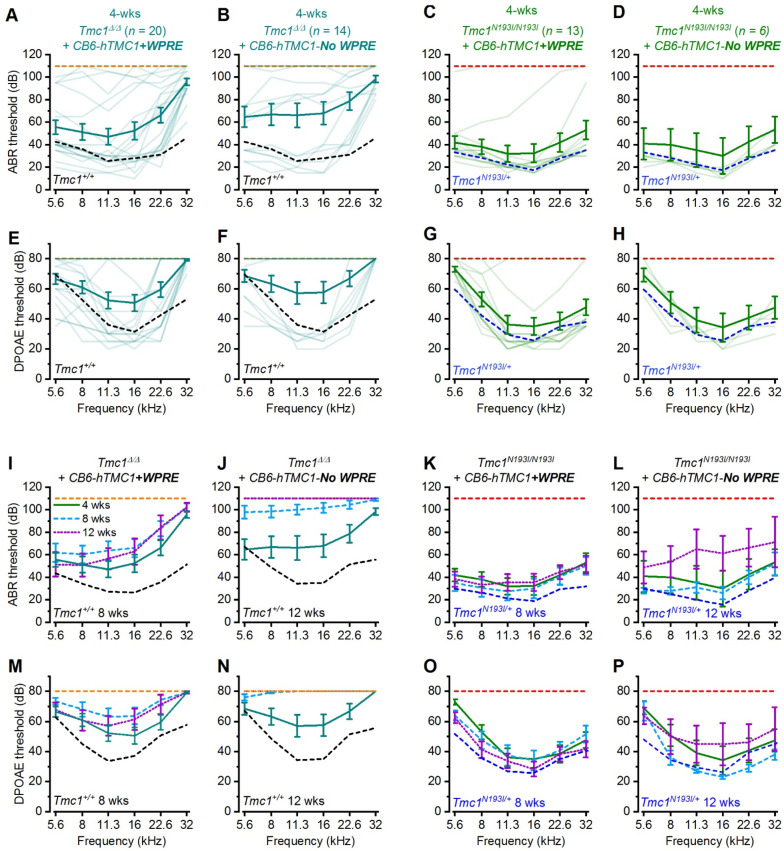
Deaf mice recover broad spectrum auditory capacity with AAV2/9-PHP.B*-CB6-hTMC1* gene therapy. ABR (**A**–**D**) or DPOAE (**E**–**H**) thresholds as a function of frequency of *Tmc1^Δ/Δ^* (**A**,**B**) (**E**,**F**) and *Tmc1^N193I/N193I^* (**C**,**D**), (**G**,**H**) mice injected at P1-2 with *CB6-hTMC1 + WPRE* (**A**,**C**,**E**,**G**) or *CB6-hTMC1 (No WPRE)* (**B**,**D**,**F**,**H**) at 4 weeks of age. Mean ± S.E.M. thresholds shown in bold traces and individual recording are shown as lighter traces. Mean ± S.E.M. ABR (**I**–**L**) or DPOAE (**M**–**P**) thresholds versus frequency for injected 4-, 8-, and 12-week-old mice. Dashed lines indicate mean ABR or DPOAE thresholds for uninjected control mice: *Tmc1^Δ/Δ^* (orange), *Tmc1^+/+^* (black) (4-week-old: (**A**,**B**,**E**,**F**); 8-week-old: (**I**,**M**); 12-week-old: (**J**,**N**)) and *Tmc1^N193I/N193I^* (red), *Tmc1^N193I/+^* (blue) (4-week-old: (**C**,**D**,**G**,**H**); 8-week-old: (**K**,**O**); 12-week-old: (**L**,**P**)).

**Figure 4 biomolecules-12-00914-f004:**
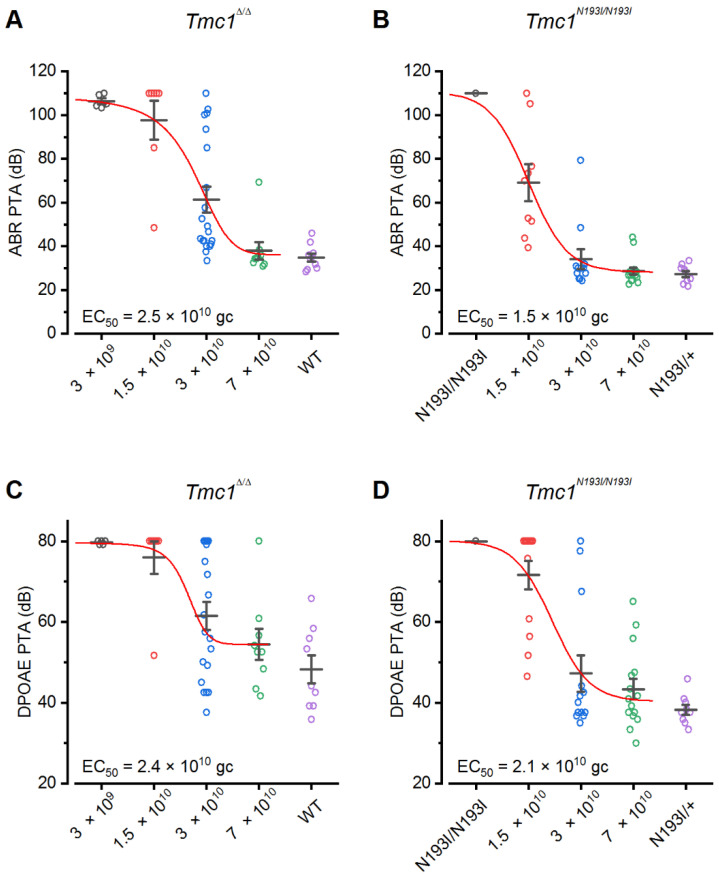
Dose–response curve of the AAV2/9-PHP.B*-CB6-hTMC1 + WPRE* gene therapy construct. Pure tone average (PTA) ABR (**A**,**B**) or DPOAE (**C**,**D**) thresholds of *Tmc1^Δ/Δ^* (**A**,**C**) or *Tmc1^N193I/N193I^* (**B**,**D**) injected with different doses of *CB6-hTMC1 + WPRE* (3 × 10^9^ gc–7 × 10^10^ gc). PTAs from uninjected controls are included in the plots for comparison.

**Figure 5 biomolecules-12-00914-f005:**
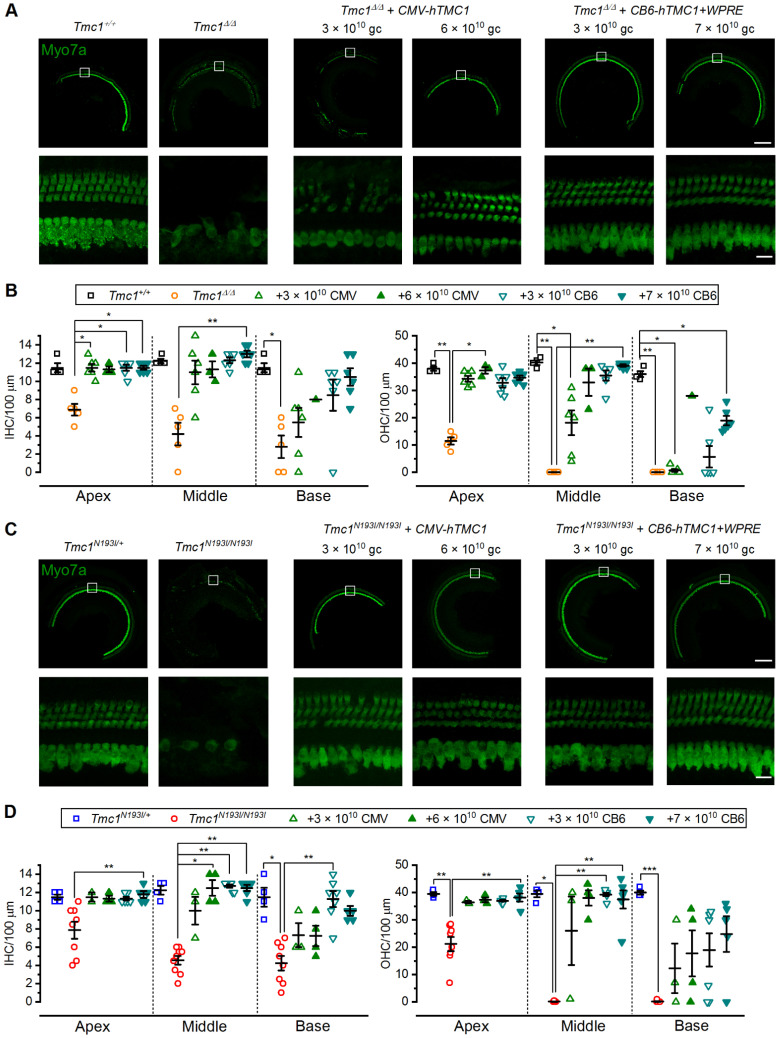
AAV2/9-PHP.B*-hTMC1* gene therapy restores hair cells counts in mutant mice. (**A**) Representative confocal images from middle cochlear sections of *Tmc1^+/+^*, *Tmc1^Δ/Δ^* and *Tmc1^Δ/Δ^* injected with *CMV-hTMC1* (3 × 10^10^ or 6 × 10^10^ gc) or *CB6-hTMC1 + WPRE* (3 × 10^10^ or 7 × 10^10^ gc) from 12-week-old mice, immunostained against myosin 7a. White boxes (above) indicate region shown at higher magnification (below). (**B**) Mean ± S.E.M. hair cell counts per 100 µm sections for IHCs (left) and OHCs (right) from untreated controls and *Tmc1^Δ/Δ^* injected mice. Individual samples are included in the scatterplots. (**C**) Representative confocal images from middle cochlear sections of *Tmc1^N193I/+^*, *Tmc1^N193I/N193I^* and *Tmc1^N193I/N193I^* injected with *CMV-hTMC1* (3 × 10^10^ or 6 × 10^10^ gc) or *CB6-hTMC1 + WPRE* (3 × 10^10^ or 7 × 10^10^ gc) from 12-week-old mice, immunostained against myosin 7a. White boxes (above) indicate region shown at higher magnification (below). (**D**) Mean ± S.E.M. hair cell counts per 100 µm sections for IHCs (left) and OHCs (right) from untreated controls and *Tmc1^N193I/N193I^* injected mice. Individual samples are included in the scatterplots. * *p* < 0.05; ** *p* < 0.01; *** *p* < 0.001; Kruskal–Wallis followed by Dunn’s multiple comparison test. Scale bars: top panels (10×) 200 µm; low panels (63×) 15 µm.

## Data Availability

The data presented in this study are available on request from the corresponding author.

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
