# Peer review of "Optimized AAV Vectors for TMC1 Gene Therapy in a Humanized Mouse Model of DFNB7/11"

_biomolecules, 2022, doi:10.3390/biom12070914_

Round 1
Reviewer 1 Report
The manuscript titled “Optimized AAV vectors for TMC1 gene therapy in a humanized mouse model of DFNB7/11”, by Marcovich and colleagues, is a very well established, thorough study of the potential of various AAV9-PHP.B constructs carrying a human TMC1 (hTMC1) sequence to revert the phenotype caused by a mutation homologous to the p.N199I mutation that causes progressive moderate-to-severe hearing loss in patients. Within the scope of this work, the authors have generated a humanized mouse model (Tmc1N193I/ N193I), since none of the available murine Tmc1 models harbour mutations identified in human patients, as well as developed an optimized viral construct where stable expression of the human gene is driven by a CB6 promoter and long-term expression of hTMC1 is achieved through the insertion of a WPRE element. It is with this viral construct that Marcovich and co-authors have registered full recovery of the hearing function in 10 out of 13 mutant animals, recovery that spans across all frequencies. This is without doubt the main finding of this work. Results from studies on different constructs and the viral titres required to observe some rescue of the mutant phenotype, as well as on the duration of the observed effects, and some preliminary evaluation on the safety of the virus are shown. Interestingly, differences are observed in the response of Tmc1D/D mice and that of the Tmc1N193I/ N193I model that suggest that the mutated protein, although not functional, must be playing some role.
The science is sound and extensive work has been carried out. The number of animals is very considerable, although some of the test groups include too few animals (n=2; even, n=1). The methodology has been very clearly described and all data are carefully interpreted, with the results being shown in detail in the Figures and Tables provided.
A major comment to the work is the absence of experiments where viral injections have been conducted at different timepoints other than P1-2. One of the main points in this story is the modelling of a human mutation that leads to progressive hearing loss and that is expected to offer a broad therapeutic window. It is unfortunate that such progressive phenotype has not been attained in the model the authors have generated, but it would have certainly been of interest to know whether the same rescue (and to the same extent, regarding parameters such as sound frequencies) would be observed had the injections been administered at later times than P1-2. Work on other Tmc1 models has already shown that treatment timing influences its outcome. Some comment addressing this issue would be desirable.
Other comments would be:
- The text is plagued by typos and formatting errors. Regarding the latter, there are paragraphs and headings of some of the sections that are below the legends of some of the figures and that show the same indentation and line spacing that the legends, differently from the rest of the text. Also, please check lines 434-435: a formatting error has occurred, and the number has been changed into a date.
- In Suppl Table 1, the section corresponding to Figure 1E: The numbers of animals indicated for the Tmc1N193I/+; Tmc2‐/‐ (n=12) and the Tmc1N193I/ N193I; Tmc2‐/‐ (n=13) have been exchanged, compared to those indicated in the text of the manuscript (lines 323 and 325).
- There has been a mistake in Suppl Figure 1: Suppl Fig 1B has been included as part of Suppl Fig 1A, and what has been indicated as Suppl Fig 1B should be labelled as Suppl Fig 1C; Suppl Fig 1C should in turn become Suppl Fig 1D in the figure (the legend is correct).
- In legend to Suppl Fig 1D (lines 644-645), it would be convenient to specify that these counts correspond to 12-week-old mice
- Suppl Table 5: The table has been cut and the last column on the right appears on the next page.
- What should be Suppl Table 6 has been indicated as Suppl Table 1. Within this Suppl Table: Why is it indicated, in the part related to Suppl Figure 1C-D (Histology) that the mice are P28? Has this work not been done on 12-week-old mice?
- Axis of the middle figure in Suppl Figure 2B: Correct RDNA to DNA
Reviewer 2 Report
Marcovich and colleagues generated a Tmc1 mouse model harboring the human p.N199I mutation, and found that homozygous mice had profound, congenital hearing loss due to loss of hair cell sensory transduction. Then the authors optimized and screened viral payloads packaged into AAV9-PHP.B capsids, and injected the vectors into the inner ears of Tmc1Δ/Δ mice and the new humanized Tmc1-p.N193I mouse model. The authors found broad spectrum, durable recovery of auditory function in Tmc1-p.N193I mice injected with AAV9-PHP.B-21 CB6-hTMC1-WPRE. ABR and DPOAE thresholds were equivalent to those of wild-type mice. As such, the authors concluded that the AAV9-PHP.B-CB6-hTMC1-WPRE 24 construct may be suitable for further development as a gene therapy reagent for treatment of humans with genetic hearing loss due to recessive TMC1 mutations.
In general, this article is clearly written and the experiments were carefully conducted. The results are promising and of practical importance by demonstrating the translational potential of gene therapy in TMC1 patients. This article would be better off by addressing the following comments:
Comments
1. Please confirm the definition of “humanized mice”. My understanding is that the p.N193I variant was introduced into the mouse Tmc1 gene in this study.
2. The results are very promising and exciting. The experiments were done in neonatal mice. Did the authors have a chance to check the effects of gene therapy in elder mice or adult mice?
3. Fig 1D. Tmc1N193I/N19 3 Tmc1N193I/N193I
4. Line 349, Figure legend. 15µ 15µm
5. Line 352, p < 0.001
6. Fig 2 & 3. Please label the 4 wks in A-H, as labeled in I-P.
7. Fig 5. Legend. Please clarify the representative p values of *, ** and ***.
Reviewer 3 Report
The manuscript "Optimized AAV vectors for TMC1 gene therapy in a humanized mouse model of DFNB7/11" provides some new insights into the use of gene therapy for TMC deficiency-related deafness. Based on the original AAV9-PHP.B vector, the best therapeutic conditions were obtained by replacing the promoter and WPRE elements and exploring the optimal dose. In addition, the authors established a humanized mouse model Tmc1N193I/N193I, which is a common mutation in human DFNB7/11, and achieved complete hearing recovery using the optimized AAV vectors in this mouse model. In the future, the AAV9-PHP.B-CB6-hTMC1-WPRE vector may be well suited to advance as a gene therapy reagent for the treatment of TMC1-related hearing loss. This paper has a lot of potential benefits, though it also has several areas of concern. Here are a few major concerns:
1. In the paper, CMV-hTMC1 was used to restore treatment for Tmc1Δ/Δ and Tmc1N193I/N193 respectively, and the results showed a better treatment effect for Tmc1N193I/N193. Please explain whether there is any statistical difference between the treatment effect of CMV-TMC1 and CMV-hTMC1 for Tmc1Δ/Δ.
2. Line 429 in the article mentions that the CB6 promoter is not as strong as CMV, so how does CB6 play a better role? Is there a difference in expression efficiency or location of the two promoters to the AAV vector?
3. In Fig3, 10 of 13 mice in the CB6-hTMC1+WPRE-treated humanized Tmc1N193I/N193I line showed complete recovery of ABRs and DPOAEs to wild-type mouse levels, including the harder-to-recover high-frequency hearing. Whereas in Fig5.D, the inner hair cells and outer hair cells survive in the base turn of Tmc1N193I/N193I mice cochlear under the same treatment condition do not fully recover, and the recovery of their function is unknown. An explanation is requested for this mismatch between the degree of hearing and hair cell recovery.
4. The green fluorescence of the F and G plots in Fig1 is easily confused. It is better to indicate this in the figure with the same color text.
5. Please add the necessary cell marker fluorescence staining to determine the cell layer in the F plot in Fig1.
6. The middle cochlear sections of the Tmc1+/+ group in Fig5 are shorter than all the other groups, so it would be better to replace the picture with a similar length. In addition, please point out which part of the enlargement in the bottom row of figures A and C is derived from? Perhaps it can be marked with a box.
7. Page 15, line 499: there is extra punctuation in the sentence---brain, and liver, .
